

# Streamflow forecasts from WRF precipitation for flood early warning in mountain tropical areas

María Carolina Rogelis[1] and Micha Werner[1,2]

[1]UNESCO-IHE, PO Box 3015, 2601DA Delft, The Netherlands
[2]Deltares, PO Box 177, 2600MH Delft, The Netherlands

*Correspondence to:* Maria Carolina Rogelis
(c.rogelisprada@unesco-ihe.org)

**Abstract.** Numerical Weather Prediction (NWP) models are fundamental to extend forecast lead-times beyond the concentration time of a watershed. Particularly for flash flood forecasting in tropical mountainous watersheds, forecast precipitation is required to provide timely warnings. This paper aims to assess the potential of NWP for flood early warning purposes, and the possible improvement that bias correction can provide, in a tropical mountainous area. The paper focuses on the comparison of streamflows obtained from the post-processed precipitation forecasts, particularly the comparison of ensemble forecasts and their potential in providing skilful flood forecasts. The WRF model is used to produce precipitation forecasts that are post-processed and used to drive a hydrologic model. Discharge forecasts obtained from the hydrological model are used to assess the skill of the WRF model. The results show that post-processed WRF precipitation adds value to the flood early warning system when compared to zero precipitation forecasts. Although the precipitation forecast used in this analysis showed little added value when compared to climatology. However, the reduction of biases obtained from the post-processed ensembles show potential of this method and model to provide usable precipitation forecasts in tropical mountainous watersheds. The need for more detailed evaluation of the WRF model in the study area is highlighted, particularly the identification of the most suitable parameterisation, due to the inability of the model to adequately represent the convective precipitation found in the study area.

## 1 Introduction

Numerical Weather Prediction (NWP) models are fundamental to extend flood forecast lead-times beyond the concentration time of a watershed. The significant advances in NWP and computer power during the last decades have led to the generation of high resolution precipitation forecasts at the catchment scale, and quantitative precipitation forecasts (QPF) from high-resolution NWPs are increasingly used in flood forecasting systems as a result (Cluckie et al., 2006). Particularly for flashy watersheds with short times of concentration, such as those typically found in tropical mountainous watersheds; forecast precipitation is required to provide sufficient forecast lead times in support of timely warnings.

Despite the significant advances, NWP results contain noise, are contaminated by model biases, are too coarse to adequately resolve all features such as convection, and are influenced by uncertainty inherent in the initial conditions (Colman et al., 2013). Furthermore, weather forecasting in tropical mountains is highly challenging. In the tropics, local and mesoscale effects



are more dominant than synoptic influences (except for tropical cyclones). In addition, there are limitations in the availability of surface and upper air monitoring networks (Laing and Evans, 2010), which influences modelling initialisation (Cuo et al., 2011). Additionally, the formation and movement of deep convection and mesoscale convective systems is affected by orography (Colman et al., 2013).

According to Habets et al. (2004) the potential of NWP precipitation forecast to be used by hydrological models for flood forecasting is mainly affected by: (i) localisation of the events, since an error of a few kilometres can lead the precipitation being forecast in the wrong watershed; (ii) timing of the events, since the response of the basin depends on previous events and on the timing of the present event; and (iii) precipitation intensity. This is especially true in flash flood prone watersheds, typical of tropical mountainous areas where location errors that are considered small at meteorological level can lead to a flood
event in a watershed to be completely missed (Vincendon et al., 2011).

In order to address the uncertain nature of NWP, ensemble prediction systems (EPS) have been developed (Demeritt et al., 2007). In contrast to deterministic systems that produce one prediction, EPS produce a suite or ensemble of predictions to reflect uncertainty, providing the capability to transform predictions into a probability distribution function (Leutbecher and Palmer, 2008; Demeritt et al., 2007). As EPS rainfall predictions often exhibit greater skill than deterministic predictions, the
hope is then that EPS products will increase the skill and time horizon of flood forecasts (Demeritt et al., 2010).

The use of NWP ensembles to drive flood forecasting systems has increased and is a relevant research topic (Cloke and Pappenberger, 2009). Currently many flood forecasting centres use ensemble prediction systems (EPS) for representing uncertainty, but most of these are in Europe, Canada, The United States and Australia (Demeritt et al., 2007). Experience of this kind of system in developing countries is very limited (Fan et al., 2014).

Rainfall forecasts provided by NWP require post-processing to correct bias and to reliably quantify uncertainty (Robertson et al., 2013). Several approaches have been used to produce ensemble rainfall forecasts by post-processing raw numerical weather prediction (NWP). Robertson et al. (2013) present a method that uses a simplified version of the Bayesian joint probability modelling approach to produce forecast probability distributions for individual locations and forecast lead times; Theis et al. (2005) propose a methodology based on the hypothesis that some probabilistic information about a precipitation
forecast at a certain time and location can be derived from its spatio-temporal neighbourhood in the model precipitation field. A set of forecasts is extracted from the spatio-temporal neighbourhood of a point and used to derive a probabilistic forecast at the central point of the neighbourhood; Bremnes (2004) proposes a method to produce probabilistic forecasts in terms of quantiles from NWP output using probit regression and quantile regression; Clark et al. (2004) use a two-stage approach that includes logistic regression and ordinary least squares regression to generate precipitation and temperature ensembles. Probabilistic
forecasts of precipitation are especially challenging since precipitation has a mixed discrete-continuous probability distribution (Cortinas et al., 2002). In this study, a simple two-stage approach is used to produce precipitation ensembles from WRF forecasts, based on probit regression and quantile regression.

The assessment of the value of forecasts is a crucial issue. Verification is essential for the understanding of the abilities, weakness and value of forecasts, which leads to improving the forecast system. Several scores are proposed in literature to
asses the quality of forecasts as well as skill scores aimed at quantifying the relative accuracy of a forecast with respect to



a reference forecast (Wilks, 2011) providing an estimation of its added value. Added value is often more important than a measure of skill (Jolliffe and Stephenson, 2012).

The objective of this paper is to assess the potential of NWP for flood early warning purposes in tropical mountainous watersheds, and the possible improvement that bias correction can provide. The main contribution of the research is focused on the assessment of the potential of WRF post-processed precipitation forecasts in providing skilful flood forecasts in watersheds characterized by rough topography and a tropical environment, where experience with precipitation forecasts, particularly ensemble approaches, has been very limited.

The study area is the upper part of a high tropical mountainous watershed located in Bogotá (Colombia). This corresponds to a Páramo, which is a tropical high-mountain ecosystem, characterised by soils with a high water storage capacity and high conductivity. The hydrologic behaviour of páramo is complex, and there are still major gaps in knowledge (Sevink, 2011; Reyes, 2014; Buytaert et al., 2005, 2006). Additionally, hydrometeorological data are scarce.

In Colombia, NWP models forecasts are provided by the Instituto de Hidrología, Meteorología y Estudios Ambientales (IDEAM), the national hydro-meteorological institute. The raw output of these models is deterministic. In this study, the operational WRF model that is used by IDEAM for daily weather forecasts is used to provide precipitation forecasts that are post-processed and used to drive a hydrologic model of the study area. The discharges obtained from the hydrological model are used to assess the skill of the WRF model.

The paper is structured as follows: Section 2 describes the methods and data used for the assessment of the potential of the WRF model. In Section 3 the results are presented, including the bias correction of precipitation and the verification of precipitation and discharge forecasts. Section 4 discusses the results. Finally, the conclusions are presented in Section 5.

## 2 Methods and data

### 2.1 Study Area

The Tunjuelo river basin is located in the south of the city of Bogotá (see figure 1). Its area is approximately 380 km$^2$ and the upper basin corresponds to a páramo area. The upper basin is composed of three watersheds, Chisaca, Mugroso and Curubital (see figure 1-c). These discharge into two reservoirs (Chisaca and Regadera) with volumes of 3.3 Mm$^3$ and 6.7 Mm$^3$ respectively. The reservoirs are operated to supply 1.2 m$^3$/s of water to the south of Bogotá. Flood waves in the urbanised lower basin are dominated by the discharge release of the two reservoirs.

The last major flood event in the basin occurred in 2002, causing the river to change its course to flow into two mining pits that since act as inline reservoirs. The peak release of the Regadera dam for that event reached 100 m$^3$/s (Rogelis, 2006), which caused flooding downstream. The current flood warning criteria include warning levels set on the water levels in the Regadera Reservoir. Forecasting of the input discharges to this reservoir, as well as to the Chisaca reservoir, which is located immediately upstream, is crucial.

The upper basin of the Tunjuelo river has a unimodal precipitation regime (rainy season April-November) (Bernal et al., 2007). The largest discharges in the upper Tunjuelo basin occur during the months May, June and July (Rogelis, 2006).





The monitoring network is shown in Figure 1. Two tipping bucket telemetric rain gauges currently operate in the upper Tunjuelo river basin and three discharge gauges are available in the three watersheds that discharge into the two reservoirs of

the upper basin with hourly records.

## 2.2  WRF model data and observed rainfall fields

The hydro-meteorological agency of Colombia, IDEAM, runs the Weather Research and Forecasting (WRF) model version WRFV3.1 for Bogotá, using the initial conditions provided by the GFS model (Ruiz, 2010). The WRF model has been used at IDEAM since 2007 to produce forecasts at national level, as well as in the Bogotá region at higher resolution (Arango and

Ruiz, 2011).

The dataset of WRF forecast used in this paper corresponds to 107 selected days when significant storms were recorded. For each of these days, forecasts with the WRF model were generated at 00:00 GTM, 06:00 GTM, 12:00 GTM and 18:00 GTM. This choice of storms ignores days when the WRF model would have delivered false-alarms, focusing exclusively on the days when high precipitation was effectively measured.

The storms typically occur in the afternoon, thus the WRF forecast have a lead time in a range between 0 and 18 hours. The WRF model comprises three nested domains, centred in Bogotá. The coarsest domain covers most of the Colombian National territory with a spatial resolution of 15 km; the intermediate domain covers mainly the central and eastern Andean cordilleras with a spatial resolution of 5 km; and the finest domain covering Bogotá only has a spatial resolution of 1.67 km (Arango and Ruiz, 2011). The parameterisation of the model corresponds to that used by IDEAM for its routine forecasts (Arango and Ruiz,

2011). The lead time of the WRF model used operationally is 72 hours, though in this paper only the first 48 hours lead time was considered.

Observed hourly precipitation data was available from the tipping bucket rainfall gauges. These were used to produce rainfall fields through inverse distance weighing interpolation (IDW).

Both WRF forecast rainfall fields and IDW rainfall fields were transformed to time series of mean average precipitation for

the Chisacá, Mugroso and Curubital watersheds (see Figure 1).

## 2.3  Methodology

The hydrology of the upper watersheds of the Tunjuelo river was modelled using the TOPMODEL (Beven and Kirkby, 1979). This was calibrated and compared with two other hydrological models by Rogelis Prada (2016), and found to provide a realistic representation of hydrological processes and good performance in the páramo watersheds. Furthermore, the preference of

TOPMODEL for páramo areas has been reported previously by other authors (Buytaert and Beven, 2011). The hydrological models were forced with precipitation input obtained from inverse distance interpolation (IDW) of hourly rainfall obtained from gauges up to the time of start of the forecast (T0). From T0 the model was forced using four different rainfall forecasts; a) zero rainfall forecasts; b) raw forecasts from the WRF model; c) bias corrected WRF forecasts; d) and precipitation forecast ensembles obtained from the post processing of the WRF model.



Both precipitation forecasts and streamflow forecasts were verified through the use of standard skill scores and rank histograms. In the case of streamflow forecasts obtained from the WRF model, these were compared with a reference streamflow time series generated using the precipitation forecast equal to zero to assess the added value of the WRF forecasts. The procedures for the production of the different rainfall forecasts and the verification process are described in the following sections.

### 2.3.1 Generation of Precipitation Forecasts

The precipitation forecast to drive the models were generated under four strategies:

(a) Zero rainfall forecasts: After T0 values of zero precipitation are used to drive the TOPMODEL.

(b) Raw forecasts from the WRF: the only pre-processing of the WRF forecasts is the sampling of the grids to obtain the hourly mean areal precipitation for each watershed. No post-processing is applied.

(c) Bias correction of WRF: The time series of mean areal precipitation obtained from the WRF model are bias corrected through Distribution-Based Scaling - DBS (Yang et al., 2010). As reference values the time series of mean areal precipitation obtained from IDW are used, and the correction is carried out for each lead-time of the WRF model. The DBS approach uses two steps: (1) Correction of the percentage of wet time steps. A precipitation threshold of 0.1 mm (rainfall gauge accuracy) was used, below which a time step is considered to be dry and (2) Transformation of the remaining precipitation to match the observed frequency distribution. The gamma distribution is used to describe the probability distribution function of precipitation intensities given its ability to represent the asymmetrical and positively skewed distribution of precipitation. The distribution parameters are estimated using maximum likelihood estimation. To capture the main properties of normal precipitation, as well as of the extremes, the precipitation distribution is divided into two partitions separated by the 95th percentile. The resulting distribution corresponds to a double gamma distribution. The two sets of parameters are used to correct the WRF model outputs.

(d) Bias correction and generation of ensembles through post processing of the rainfall forecast from the WRF model: The reference and forecast time series were organised according to lead time and a two stage post processing model was applied to reflect the intermittent nature of rainfall (Rene et al., 2012; Clark et al., 2004). The first stage corresponds to the probit model (Scardovi, 2015; Kleiber et al., 2012; Bremnes, 2004) to simulate the probability of occurrence of precipitation. The second stage considers the probability distribution of the precipitation amount, given its occurrence, for which quantile regression was used. The mean precipitation time series are first disaggregated into a time series of occurrence (1 = wet time step and 0= dry time step) and precipitation amounts (wet time steps only). The time series of occurrence is used as the response variable for the probit regression model, and the time series of precipitation amounts is used as the response variable for the quantile regression model.





In the probit regression model, given binary observations of precipitation occurrence $y_1,...y_n$, and $x_{i1},...x_{ik}$ covariates associated with the ith response, the probability that $y_i = 1$, $p_i$, is written as (Albert, 2009):

$$p_i = P(y_i = 1) = \Phi(x_{i1}\beta_1 + ... + x_{ik}\beta_k) \tag{1}$$

where $\beta = (\beta_1,...,\beta_k)$ is a vector of unknown regression coefficients and $\Phi()$ is the cumulative density function of a standard normal distribution. If we assume a uniform prior for $\beta$, then the posterior density is given by:

$$g(\beta \mid y) \propto \prod_{i=1}^{n} p_i^{y_i}(1-p_i)^{1-y_i} \tag{2}$$

The binary response corresponds to the occurrence $y_i = 1$ or non-occurrence of rain $y_i = 0$. A latent variable $Z_i$ is introduced in such a way that if $Z_i$ is positive for $y_i = 1$ and $Z_i$ is negative for $y_i = 0$. This latent variable is related to the k covariates by the normal regression model:

$$Z_i = x_{i1}\beta_1 + ... + x_{ik}\beta_k + \epsilon_i \tag{3}$$

where $\epsilon_1, ..., \epsilon_n$ are a random sample from a standard normal distribution. Then:

$$P(y_i = 1) = P(Z_i > 0) = \Phi(x_{i1}\beta_1 + ... + x_{ik}\beta_k) \tag{4}$$

This can be considered as a missing data problem where there is a normal regression model of latent data $Z_1,...Z_n$ and the observed responses are missing or incomplete and can only be observed if $Z_i > 0(y_i = 1)$ or if $Z_i \leq 0(y_i = 0)$.

In order to avoid distributional assumptions, quantile regression (QR) will be used to describe the probability of the amounts of precipitation given its occurrence (Bremnes, 2004). Let $r_1,...r_{n*}$ denote observed precipitation amounts of cases with observed precipitation above a given lower threshold and $z_1,...z_{n*}$ corresponding predictor values where $z = (z,...,z)^T$. For linear quantile functions (Bremnes, 2004):

$$q_\theta(z_i;\beta) = \beta_0 + \sum_{k=1}^{K} \beta_k z_{ik} \tag{5}$$

an estimate of the $q_\theta(z_i;\beta), 0 < \beta < 1$ quantile, is obtained by solving the following minimization problem with respect to $\beta$:

$$\underset{\beta}{\arg\min} \sum_{i=1}^{n*} p_\theta(r_i - q_\theta(z_i;\beta)) \tag{6}$$





where the function $q_\beta(*)$, is defined in terms of the absolute deviation of residuals (u) by:

$$p_\theta(u) = \begin{cases} u\theta & if\ u \geq 0 \\ u(\theta - 1) & otherwise \end{cases} \tag{7}$$

If several quantiles are of interest, the minimization must be repeated for each quantile. A potential problem with using QR for the derivation of multiple conditional quantiles is that quantiles may cross, yielding predictive distributions that are not monotonously increasing, as a function of increasing quantiles (López López et al., 2014). In the present research study, the technique proposed by Muggeo et al. (2013), and implemented in the package quantregGrowth developed in the R environment (R Development Core Team, 2010) is used. This technique estimates nonparametric growth charts via quantile regression.

Quantile curves are estimated via B-splines with a quadratic penalty on the spline coefficient differences, and non-crossing and monotonicity restrictions are set to obtain plausible estimates.

    Two configurations were tested: a) Quantile regression applied to the raw precipitation data; b) Quantile regression on the data transformed into normal domain through normal quantile transformation (NQT) (Bogner et al., 2012). For configuration b) the time series of observed precipitation and forecast precipitation are transformed into the normal domain. After the derivation

of the quantiles, the variables are back-transformed into original space. The rationale for using the transformation is that the joint distribution of transformed time series appears to be more linear, and can thus be better described by linear conditional quantiles (López López et al., 2014).

    Back-transformation is, however, problematic if the quantiles of interest lie outside of the range of the empirical distribution of the untransformed variable in original space. To address this issue linear extrapolation in the tails of the distribution was

used (López López et al., 2014).

    To generate the probabilistic forecast of the occurrence of precipitation, the latent variable in the probit model is first sampled. The random value u is sampled from a uniform distribution. If $Z_i < u$ no precipitation occurs, if $Z_i \geq u$ precipitation is set to occur and the amount is computed with the quantile regression model. Given the occurrence of precipitation, the quantile regression model corresponding to the lead-time under consideration is used to obtain the quantiles of the conditional distribu-

tion. The inverse cumulative function is approximated by a cubic spline and subsequently 50 uniform random numbers in the range 0-1 are generated and used to sample corresponding precipitation values from the inverse cumulative function.

### 2.3.2   Verification of forecasts

Deterministic forecasts of precipitation and discharge were verified using the Mean Absolute Error (MAE), the Mean Squared Error (MSE), the Mean Error (ME) and the skill score (SS) based on the MSE using as reference forecasts provided by climatological values (subscript Clim in Equation 11 and Equation 12) (Wilks, 2011) and zero forecasts (subscript 0 in Equation 13 and Equation 14). The corresponding formulas are shown in Equation 8 to Equation 14, where $(y_k, o_k)$ is the $k_{th}$ of n pairs





of forecast and observed values at a particular lead time. In the case of discharge, the discharge forecast obtained from the

hydrological model driven with a precipitation input equal to zero, was used as reference.

$$MAE = \frac{1}{n} \sum_{k=1}^{n} |y_k - o_k| \tag{8}$$

$$MSE = \frac{1}{n} \sum_{k=1}^{n} (y_k - o_k)^2 \tag{9}$$

$$ME = \frac{1}{n} \sum_{k=1}^{n} (y_k - o_k) = \bar{y} - \bar{o} \tag{10}$$

$$MSE_{clim} = \frac{1}{n} \sum_{k=1}^{n} (\bar{o} - o_k)^2 \tag{11}$$

$$SS_{Clim} = \frac{MSE - MSE_{Clim}}{0 - MSE_{Clim}} = 1 - \frac{MSE}{MSE_{Clim}} \tag{12}$$

$$MSE_0 = \frac{1}{n} \sum_{k=1}^{n} (0 - o_k)^2 \tag{13}$$

$$SS_0 = \frac{MSE - MSE_0}{0 - MSE_0} = 1 - \frac{MSE}{MSE_0} \tag{14}$$

In order to assess the performance of the probit model constructed to model the occurrence of precipitation from the WRF
model, the ROC (Relative Operating Characteristic) diagram was used. This is a discrimination-based graphical forecast ver-

ification display (Wilks, 2011). For perfect forecasts the ROC curve consists of two line segments coincident with the left
boundary and the upper boundary of the ROC diagram. At the other extreme of forecast performance, random forecasts consis-
tent with the sample climatological probabilities, the ROC curve will consist of the 45 degrees diagonal connecting the points
(0, 0) and (1,1). ROC curves for real forecasts generally fall between these two extremes, lying above and to the left of the
45 degrees diagonal. Forecasts with better discrimination exhibit ROC curves approaching the upper-left corner of the ROC

20   diagram more closely, whereas forecasts with very little ability to discriminate the event exhibit ROC curves very close to the
diagonal (Wilks, 2011). To summarize the ROC diagram the area under the ROC curve was used (Wilks, 2011).

Regarding the quantile regression models, the Pseudo R-Square measure (Koenker and Machado, 1999) was used to compare
the models with normal quantile transformation and the ones based on raw data. The Pseudo R-Square measure was proposed




by Koenker and Machado (1999) as a goodness of fit indicator for quantile regression by comparing the sum of squared distances for the model of interest with the sum of squared distances between the observed and the fitted values that would be obtained if only the intercept term is included in the model (Hao and Naiman, 2007).

Ensemble forecasts of both precipitation and discharge were verified using rank histograms. Rank histograms allow evaluating whether a collection of ensemble forecasts satisfy the consistency condition (Wilks, 2011). Rank histograms are constructed by accumulating the number of cases over space and time when the verifying analysis falls in any of m+1 intervals, where each of the m +1 intervals is defined by an ordered series of m ensemble members, including the two open ended intervals (Jolliffe and Stephenson, 2012). Reliable or statistically consistent ensemble forecasts lead to a rank histogram that is close to flat (Jolliffe and Stephenson, 2012).

In order to assess the probabilistic forecasts of discharge obtained from the WRF precipitation ensembles, the Continuous Ranked Probability Skill Score (CRPSS) was used. The CRPSS is given by (Hersbach, 2000; Alfieri et al., 2014):

$$CRPSS = \frac{\overline{CRPS_{ref}} - \overline{CRPS_{forecast}}}{\overline{CRPS_{ref}}} \tag{15}$$

where:

$$CRPS = \int\limits_{\infty}^{-\infty} \left[F(y) - F_0(y)\right]^2 dy \tag{16}$$

and

$$F_0(y) = \begin{cases} 0, & y < observed\ value \\ 1, & y \geq observed\ value \end{cases} \tag{17}$$

$F(y)$ is a stepwise cumulative distribution function (cdf) of the ensemble of each considered forecast. CRPSS ranges between 1 and $-\infty$. A CRPSS of 1 indicates a perfect forecast. Forecast ensembles are only valuable when $CRPSS > 0$, this is when the forecasts perform better than the reference. The discharge obtained from forecast precipitation equal to zero will be used as reference. Since this is a deterministic forecast, the $CRPS_{ref}$ corresponds to the mean absolute error of all forecast and observed pairs (Hersbach, 2000).

## 3 Results

### 3.1 Bias correction of precipitation forecasts through DBS

Figure 2 shows the empirical cumulative distribution functions (ECDF) of the raw WRF data, the observed precipitation obtained from IDW interpolation, and the bias corrected WRF data for the Mugroso watershed, results for the other two watersheds were similar and not shown here for brevity. The raw WRF and IDW time series were obtained from the corresponding





rainfall fields as the mean areal precipitation calculated in the three watersheds of analysis by sampling the raster layers with
25 the polygons of the watersheds.

The ECDF of raw WRF precipitation shows that both overestimation and underestimation of precipitations occurs. Underestimation is very noticeable for lead times up to 2 hours for values in the upper percentiles. In general for lead times of 3 hours and higher, overestimation of high precipitation values occurs.

## 3.2 Quantile regression model

Figure 3 shows the Pseudo R-Square for all quantile regression models for the Mugroso watershed and for the coarsest domain
(resolution of 15 km), the results are similar for the other watersheds and domains. The R-Square values are pooled for lead times up to 12 hours. 12 hours was chosen as a first estimate of the usable lead-time for the forecast, since NPW models describing local-regional convective systems and orographic processes have been observed to significantly decrease their skill at lead times beyond 12-48 hours (Cuo et al., 2011). The figure shows that the values of the Pseudo R-Square up to the 50th percentile are higher when the normal quantile transformation is used, thus indicating a better fit of the quantile regression
model. For higher percentiles the Pseudo R-Square of the models with normal quantile transformation is in approximately the same range of the raw data or lower. This means that for percentiles above the 50th percentile the fit of the quantile regressions is better for the raw data than for the normal transformed data.

## 3.3 Verification of precipitation forecasts

## 3.4 Verification of deterministic precipitation forecasts and ensemble mean

Figure 4 shows the performance measures showed in Equation 8 to Equation 14. The performance of all the WRF domains are shown for the three watersheds (Chisaca, Mugroso and Curubital). From top to bottom the figure shows the ME, the MAE and the MSE. The lowest two figures for each watershed show the skill score based on MSE using climatology as a reference, and the skill score using zero precipitation as a reference.

The ME of the raw WRF forecasts shows that in the three watersheds for lead times between 1 and 5 hours the precipitation
amounts are overpredicted (positive ME), while after 5 hours the ME shifts to negative values, indicating underprediction. The bias corrected WRF forecasts show values of ME very close to zero, which could be expected given the procedure applied (see Figure 4). In the case of the mean of the WRF ensemble without transformation, the ME is close to zero for lead times up to 6 hours. Beyond this lead-time there is a tendency to under-predict precipitation. The behaviour of the ME for the mean of the ensemble with normal quantile transformation is very similar to that when the transformation is not applied.
The highest values of MAE (see Figure 4) are obtained from the DBS bias corrected forecast for lead times larger than 6 hours, while the raw WRF forecasts exhibit the highest values at the shorter lead times. In the case of the MSE, the highest values are obtained from the bias-corrected WRF forecasts at lead times of 2 and 8 hours. The increase in MSE when bias correction is applied is mainly attributed to the influence of a single (observed) high precipitation event that was not forecast by the WRF model. These intense precipitation events being missed in the forecasts cause all WRF values higher than the




95th percentile to be significantly increased through the bias correction procedure, increasing their square difference with the observed value. These extremes are mainly present at lead times of 2 and 8 hours due to extreme precipitation events in this area typically occurring in the mid-afternoon, some 2 to 8 hours after the start of the 18:00 GTM and 12:00 GTM WRF forecast runs (the time zone of the area is UTC-5).

For both MSE and MAE, the best performance is obtained from the ensemble mean of the WRF forecasts. This is the case both with and without normal quantile transformation (see Figure 4), which provide very similar performance. Regarding the skill score based on the MSE using the climatology as reference (see Equation 11 and Equation 11) negative values are obtained

for all forecasts, with the bias corrected WRF and the raw WRF forecasts showing the worst skill. However, in the case of the ensemble mean, both for the normal-quantile transformed and raw ensembles, the values are very close to zero, albeit still negative. When zero precipitation is used as reference, the behaviour of the score is very similar to SSclim. However, most values for the ensemble mean forecast are positive, both with raw and with normal quantile transformed data, with a maximum value of 0.16. This shows that the ensemble mean forecasts provide the best results in comparison to the other forecasts. In

the case of the climatology being used as reference, they are approximately as good as the reference and compared to zero precipitation as reference; they provide an improvement up to 16% over the reference.

Given that a single extreme storm which was entirely missed by the WRF forecast has a significant influence in the results shown in Figure 4, (resulting in the poor performance at lead times of 2 and 8 hours), the accuracy metrics were recalculated from the data set, but now excluding this storm. The results are shown in Figure 5 only for the MSE, since the other metrics

are less affected by the exclusion of the extreme storm. In Figure 5 the reduction of the MSE is highly noticeable in all the forecasts (see Figure 4 in comparison with Figure 5 ), and the lowest values are exhibited by the ensemble mean of both raw and normal quantile transformed data, which show very similar behaviour.

This analysis clearly shows the sensitivity of performance measures based on squared residuals, such as MSE, to a single event with high precipitation being missed in the WRF forecasts.

**3.5   Verification of deterministic discharge forecasts and ensemble mean**

The results for discharge using the precipitation forecasts are shown in Figure 6. The structure of the figure is similar to that of Figure 4, although the skill score is now determined using the climatology as reference, as the zero precipitation forecast is considered as one of the forecasts. The ME shows that the raw WRF forecasts overestimate the discharges up to a lead time of some 8 hours, while for larger lead times underestimation occurs. When the bias corrected WRF forecasts are used,

the ME is close to zero, slowly increasing with lead-time. The use of the mean of the ensembles with and without normal quantile transformation produces ME close to zero up to lead times of 4 hours, beyond which underestimation occurs (negative values of ME). In terms of the MSE the mean of the ensemble with and without normal quantile transformation produce the smallest values for all lead-times, which is reflected in the SSclim, where they produce the highest values, although there is little improvement in the skill over that obtained from using a zero precipitation series as forecast.

Figure 7 shows the MSE calculated using the stream flow simulations, but now excluding the storm with the highest precipitation that was missed by the WRF model. Only the MSE is shown, since the other metrics are less affected by the exclusion of





the extreme storm. This shows a significant improvement in comparison with Figure 6, particularly for the DBS bias corrected data.

From the results shown in Figure 4 to Figure 7, the difference in performance for the three domains of the WRF model is not significant, both in the precipitation and in the discharge forecasts.

### 3.6 Verification of probabilistic forecasts

Figure 8 shows the area under the ROC curve (AROC), showing the skill of the forecasts to discriminate between the occurrence

and non-occurrence of precipitation. Scores are shown for the three watersheds (Chisaca, Mugroso and Curubital in Figure 2); for the three domains of the WRF model for lead times up to 12 hours; and for a precipitation forecast of zero. The behaviour of the AROC is similar in the three watersheds, with the highest values found in the first two hours, dropping to a value close to 0.5 after a lead time of about 5 hours. The AROC values for a forecast of zero precipitation is very close to 0.5 for all lead times, thus is almost equal to random forecasts consistent with the sample climatological probabilities (Wilks, 2011).

This shows that the forecast have some skill at short lead times in comparison to both zero precipitation forecasts and random forecasts, although that skill is limited.

The rank histograms of the precipitation ensembles are shown in Figure 9. The behaviour for all domains and all lead times is similar (Figure 9 shows lead-times up to six hours corresponding to the left column of the graph). The rank histogram is approximately uniform from rank 1 to rank 5 and from rank 6 to rank 10 there is an increase in the frequency of the IDW

precipitation values falling in these higher ranks. Similar behaviour in all rank histograms was observed when the storm with the highest precipitation missed by the WRF model was excluded. This suggests that the WRF ensemble is slightly underforecasting.

Figure 10 shows the rank histograms for the discharges obtained from the precipitation ensemble using the TOPMODEL. The rank histograms show approximate uniformity mainly for lead times up to 3 hours. For longer lead times a slight under

dispersion is observed.

Figure 11 shows the CRPS for the discharge ensembles, the MAE for the forecast produced by using zero as precipitation forecast and the corresponding CRPSS. The CRPSS exhibits values in the range 10.5-21% (see Figure 11 ) with most values in a range of 14-16%. The same behaviour is observed in the three watersheds. The comparison between normal quantile transformed quantile regressions and the quantile regressions with raw data again shows that the difference between the two is

negligible.

### 3.7 Discussion

#### 3.7.1 Evaluating precipitation forecasts from the WRF model

The comparison of ECDF of the raw WRF forecasts and the ECDF of the precipitation obtained from IDW interpolation shows that the WRF model tends to over predict precipitation at the short lead times, which are most relevant to flash flood forecasting

in small watersheds. This behaviour has been observed in other tropical areas (Mourre et al., 2015). While the details of the



implementation of the WRF model are out of the scope of this study, possible causes of the precipitation error found by other authors include; errors in the lateral boundary conditions (Ochoa et al., 2014); poor representation of the topography (Ochoa et al., 2014); and choice of convective treatment, microphysics and planetary boundary layer (Jankov et al., 2005).

Figure 12 shows the precipitation obtained from IDW interpolation compared with the precipitation forecast of the WRF model for the highest resolution domain (1.67 km) and all the storms considered in this analysis. It can be observed that there is a lack of correlation between both data series. Most WRF values are higher than the IDW values, and there are individual cases where the WRF model did not detect the occurrence of high precipitation. Missing or underpredicted forecasts of intensive

precipitation have also been observed by other authors (Kryza et al., 2013; Liu et al., 2015), which may suggest the need for an improvement in model parameterisation and data assimilation.

No significant differences among three domains of the WRF model were found in the behaviour of the scores for the tests carried out in this study. Other applications of NWP (e.g. Roberts et al. (2009)) show that finer resolutions are capable of producing more accurate predictions, and that physics configuration, resolution and initial conditions highly influence the

WRF model performance (Kryza et al., 2013). The similarity of results regardless of resolution found in this study may be also related to deficiencies in the parameterisation of the model, or to the inability to sufficiently resolve the topography. A more detailed review of the WRF model would be required to reveal possible deficiencies, and to suggest improvements.

The WRF model has been shown to be highly sensitive to the parameterisation of cumulus and microphysical processes (Remesan et al., 2014; Rama Rao et al., 2012). Parameter sensitivity is generally dependent on local conditions (Di et al.,

2014), and the best configuration varies with time and rainfall threshold (Jankov et al., 2005). A step forward would be to test various model parameterisations, assessing their sensitivity and searching for an optimal or a set of optimal configurations that help to improve the prediction of intense convective precipitation. Observing systems such as radars, gauges and satellites for data assimilation in NWP analysis could further improve the NWP forecasts (Rossa et al., 2011; Cuo et al., 2011; Yucel et al., 2015; Liu et al., 2015).

The application of the DBS approach has been reported previously to considerably reduce the differences in rainfall frequency between the observations and forecasts (Yang et al., 2010). This reduction of differences was found in this study, as shown in Figure 12. However, where high precipitation values were observed but not forecasted in a single extreme event, the bias correction had the effect of reducing overall performance, particularly when measured by the MSE, since this is more sensitive to large residuals. This behaviour suggests that the DBS bias correction can introduce undesirable effects, which limit

its effectiveness (Ehret et al., 2012). Similar behaviour was observed with the mean of the ensemble, although with less impact on the performance measures in comparison to the DBS approach. Significant difference between the ensemble means obtained form raw and normal quantile transformed data were not found.

These biases are the result of the inability of the WRF model to adequately represent convective precipitation, failing to correctly predict when large events occur, or where it occurs. This has also been observed by other authors. For instance,

(Verkade et al., 2013) found that post-processing does not improve on all qualities at all lead times and at all levels of the verifying observations. The cause is rooted in the impossibility of the post-processing approaches to replace adequate model representation of physical processes (Haerter et al., 2011). Haerter et al. (2011) found that the conditions on climate model





data to make the application of statistical bias correction schemes reasonable are that a realistic representation of the physical processes involved must be ensured; and that the quantitative discrepancies between the modelled and observed probability density function of the quantity at hand must be constant in time. Similarly, Chen et al. (2013) emphasize the impossibility any bias correction method being successful if there is no coherence between simulated and observed precipitation. The WRF data used in this study shows limitations in fulfilling both these conditions, which leads to the post-processed precipitation being no more skilful than the sample climatology and providing only a modest improvement in comparison to the zero precipitation

forecast. However, an important result of this analysis is that despite the limitations, the ensemble mean outperforms the DBS bias correction and seems to be less sensitive to the presence of intense precipitation events that are not forecast by the WRF model. Post-processing the deterministic forecasts to generate the ensembles is clearly more sophisticated than just correcting the mean bias (Robertson et al., 2013). During the process, different distributions are used for each raw WRF forecast, which results in an improvement of skill that outperforms the bias correction through DBS. The magnitude of the bias of the ensemble

mean is nearly always smaller than the raw forecasts, as well as of the DBS bias corrected forecasts, particularly for lead times up to 6 hours.

Regarding the verification of the precipitation ensembles, the rank histograms show that the IDW precipitation too often falls between ranks 6 and 10. This reflects overdispersion of the ensemble towards low values. This produces an underforecasting bias in the ensemble. The same behaviour of the rank histograms is observed in the ensembles obtained from quantile regres-

sion, both when the normal quantile transformation is used in the precipitation data and when raw data is used. This means that the transformation of the values does not improve the consistency of the ensemble.

The AROC of the occurrence of precipitation shows that values higher than 0.5 are obtained in all watersheds for lead times up to 5 hours, indicating that the probit model has the ability to discriminate between the occurrence or non-occurrence of precipitation up to these lead times. In all domains and watersheds, the highest values of AROC are obtained for lead-times

between 2 and 3 hours, in the range of 0.63-0.67. This means that there is some skill in the probit model to forecast occurrence of precipitation for lead times up to 5 hours, even if the skill of the amount of precipitation is low. For longer times the probit model does not have any skill, consistent with other studies where the performance significantly reduces after a few hours (Liu et al., 2015). However, even though the lead times at which the model still has skill are limited, this does imply that the current ability of NWP in tropical mountainous areas can provide a limited extension of lead time beyond the concentration time of

small, flashy, watersheds.

### 3.7.2   Evaluating discharge forecast

The comparison of Figure 6 and Figure 7 shows that the biases in the precipitation forecasts reflect in the biases of the discharge. When the single event with high precipitation that is not forecast by the WRF is excluded from the analysis, a reduction of bias in the forecast precipitation is observed, as well as a reduction in the bias of the forecast discharge. This is consistent with

the findings of other studies where the errors in bias corrected precipitation lead to amplified errors in modelled runoff (Teng et al., 2015), since in this analysis the increase in errors in the bias corrected time series amplify the errors in the simulated discharges.





The forecasts generated with the post-processed ensemble precipitation forecasts exhibit better skill than the deterministic forecasts (based on raw, bias corrected and zero precipitation). Higher skill of the ensemble mean in comparison with deterministic forecasts has also been found in other flood forecasting systems driven by the WRF model (Calvetti et al., 2014) and other NWP models (Vincendon et al., 2011).

In terms of the lead-time at which the coupled WRF-TOPMODEL forecasts provide value in these watersheds, a fist limit is provided by the skill of the WRF model to forecast the occurrence of precipitation up to 5 hours (see Figure 8). A limit

is also observed in the skill scores of the discharge forecast in Figure 6 and Figure 7. Beyond the lead-time of 2 hours, which is approximately the time of concentration of the watersheds, the skill (SSclim in Figure 6 and Figure 7) starts to drop significantly. For the discharge obtained from raw and DBS bias corrected precipitation, three and four hours are the respective lead times at which the added value is totally lost in comparison with the climatology (SSclim in Figure 6 and Figure 7 reaches zero). In the case of the ensemble means, the skill also decreases progressively after 2 hours, with some value compared to

climatology up to a lead-time of some 6 hours. Beyond this lead-time, the values remain approximately constant in a range of 0.2 to 0.5 for the three watersheds. This behaviour indicates that the use of the ensemble mean provides some skill up to a lead time of 6 hours. This is three hours longer than the lead time provided by the raw WRF forecast or the DBS bias corrected WRF forecast. A lead time of 6 hours is consistent with other flood forecasting systems in small mountainous catchments driven by NWP models (Verkade and Werner, 2011).

The hydrological model produces ensemble results with rank histograms that do not reflect the overprediction of the precipitation ensemble, with approximately uniform rank histograms mainly for the first lead times. Approximately the same shape of the rank histograms is observed for ensembles obtained from quantile regression when no transformation is used in the precipitation data and when raw data is used. The minor influence of the transformation is consistent in all the performance assessments. This may be due to the relatively low improvement of the fit of the quantile regressions when the normal quantile

transformation is used. According to Figure 3, the most significant improvement is found in the low percentiles (up to the 25th percentile). For higher percentiles, the improvement is not significant and for some percentiles degradation of goodness of fit occurs. As floods occur for higher rainfall percentiles, the similarity in behaviour is logical.

The results of the CRPSS show that forcing the hydrological model with WRF ensembles improves the forecasts in a range of 8.5-22% in comparison with a forecast produced with the hydrological model forced with zero precipitation for lead times

between 1 and 12 hours. The positive values of the CRPSS imply added value to the forecasts, albeit modest. The CRPSS obtained from the ensembles is comparable to the CRPSS values found in other areas with other NWP models, albeit in the low range. E.g. Robertson et al. (2013) found a CRPSS of 37% on average for post processed ensembles in Australia, where rainfall is predominantly produced by large-scale synoptic systems that are better predicted by NWP models. Therefore, given the high complexity of the meteorological conditions of the study area, and despite the relatively poor skill of the WRF

model in predicting precipitation amounts, the WRF model does show promise at producing a benefit in its use for flood forecasting compared to not using precipitation forecasts. This is likely due to the skill found in the probit model in predicting the occurrence of rainfall.



Besides improvements in parameterisation and data assimilation, as described previously, bias correction of stream flow could provide a further skill improvement (Yuan and Wood, 2012).

## 4    Conclusions

This paper presents the assessment of WRF forecasts produced in a tropical high-mountain watershed in Bogotá, Colombia. The WRF forecasts were used to force a hydrological model of the study area. Simulated discharges were used to assess the value of the WRF forecasts for flood early warning. Although the skill of the forecasts developed is limited, results show that the streamflow forecasts obtained from a hydrological model forced by post-processed WRF precipitation do add value to the flood early warning system, particularly when compared to zero precipitation forecasts.

The WRF model for the study area provides forecasts that are found to overpredict precipitation for the shorter lead times (up to 6 hours), and that tend to fail to forecast high intensity precipitation events. This behaviour may be due to parameterisation deficiencies, errors in boundary conditions and/or poor representation of the topography. A more detailed evaluation of the WRF model in this study area is recommended, including the assessing of other convective and microphysics schemes to identify the most suitable parameterisation. The use of satellite and soon to be available radar data may further help improve performance.

Bias correction of the WRF forecast precipitation, through Distribution-Based Scaling (DBS) was shown to significantly reduce the differences in rainfall amounts between observations and forecasts. However, the DBS corrected forecasts were found to be sensitive to failure of the WRF model to predict a small number of extreme events, resulting in a reduction of overall performance. This sensitivity was also observed with the ensemble mean forecasts, although for this post-processing technique the impact of these few missed events on the performance was lower when compared with DBS. This undesirable behaviour is the result of the inability of the WRF model to adequately represent convective precipitation, which cannot be corrected through simple post-processing.

Despite the limitations in the WRF forecasts, the ensemble mean was found to outperform the DBS bias correction, as well as being less sensitive to the presence of highly intense precipitation that are not correctly forecast by the WRF model. Although none of the precipitation forecast that were assessed in this analysis showed added value when compared to climatology, the reduction of biases obtained from the ensembles show potential of this method and model to provide usable precipitation forecasts. Although the skill of the quantitative precipitation models was found to be low, the probit model, which was applied to forecast the probability of occurrence of precipitation, based on the WRF forecast, showed that the WRF model does have some skill at short lead times (up to 5 hours) in comparison to both zero precipitation forecasts and random forecasts, although that skill is limited.

Increases in precipitation biases are reflected in the discharge forecasts. However, discharge forecasts generated with the post-processed ensembles did exhibit higher skill than deterministic forecasts (based either on raw, bias corrected and zero precipitation). Furthermore, the quality of these forecasts is better than that which could be obtained using zero precipitation as input to the hydrological model. The potentially usable lead times for discharge forecasts obtained from the WRF model



found in this analysis (5-6 hours) are in the range of lead times found in other studies with other NWP models. Despite the fact that the added value of the WRF model forecasts is modest, the results found do show there is some potential for increasing forecast skill in areas of high meteorological and topographic complexity, as well as how these can be improved.

*Competing interests.*  The authors declare that they have no conflict of interest.

*Acknowledgements.*  This work was funded by the UNESCO-IHE Partnership Research Fund - UPARF in the framework of the FORESEE project. We wish to express our gratitude to the IDEAM - Instituto de Hidrología, Meteorología y Estudios Ambientales for providing the WRF model for this analysis.





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







**Figure 1.** Study area. Service Layer Credits: Esri, DeLorme, NAVTEQ, TomTom, Intermap, increment P Corp., GEBCO, USGS, FAO, NPS, NRCAN, GeoBase, IGN, Kadaster NL, Ordnance Survey, Esri Japan, METI, Esri China (Hong Kong), swisstopo, and the GIS User Community




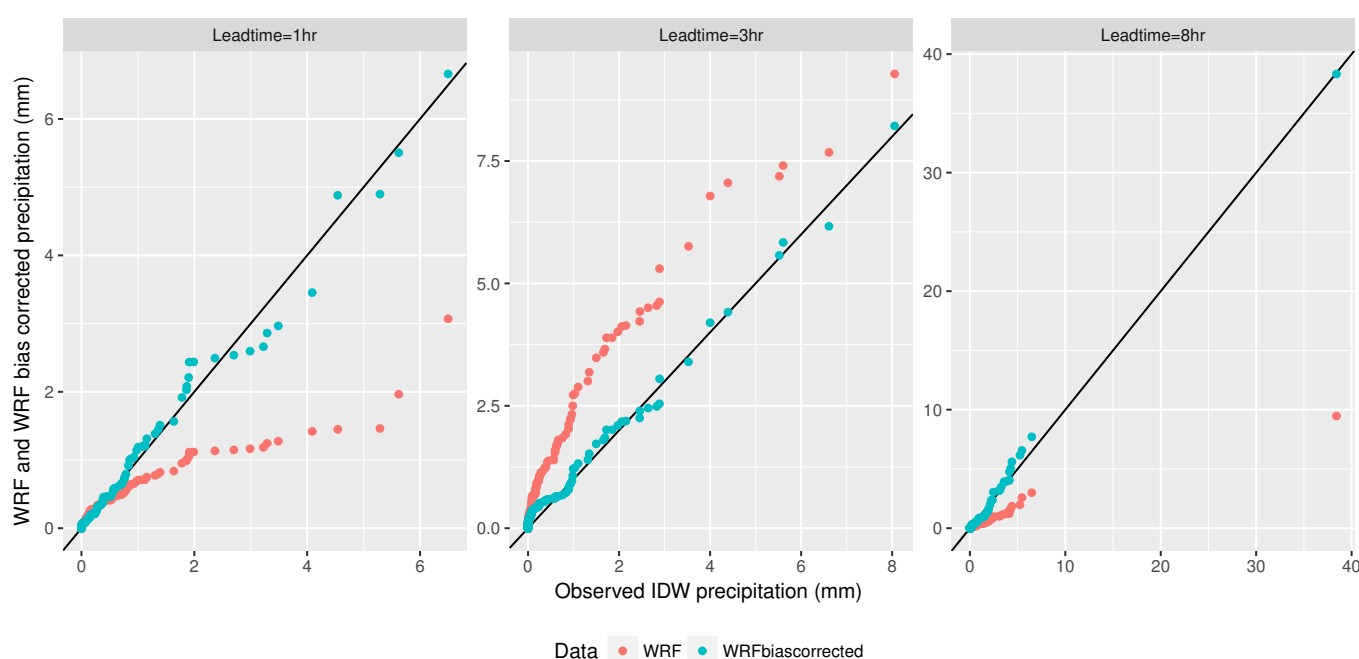

**Figure 2.** Empirical cumulative distribution function (ECDF) for the Mugroso watershed hourly precipitation for lead times up to 6 hour.



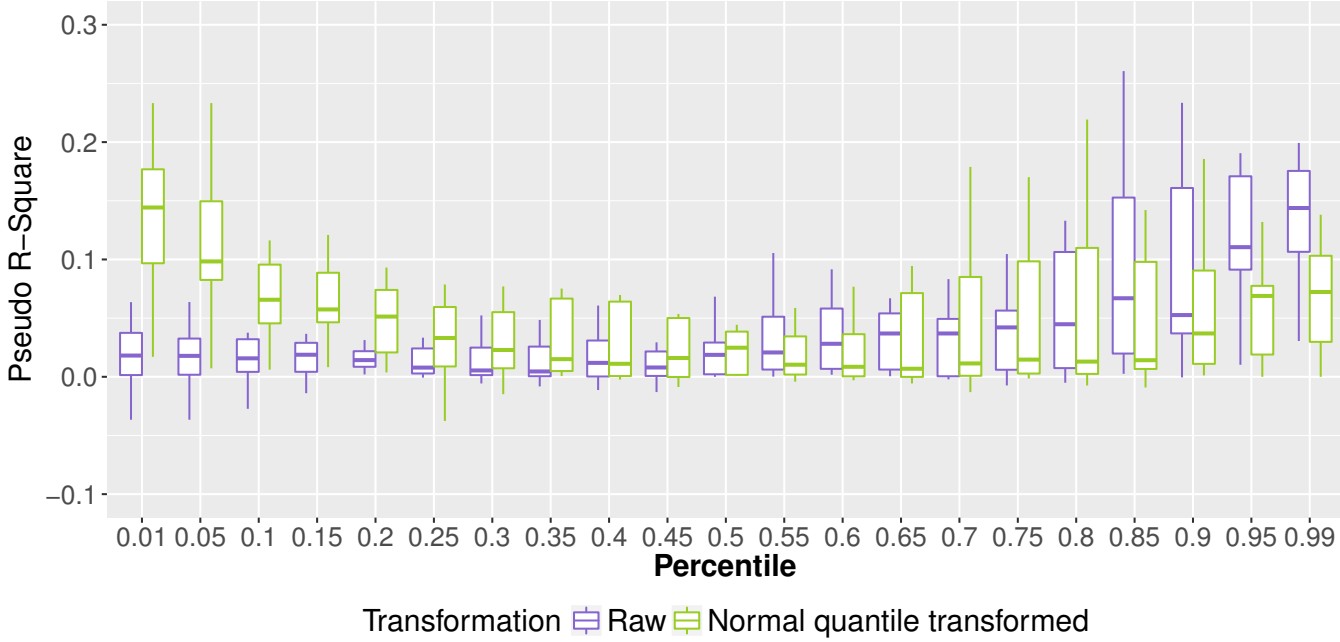

**Figure 3.** Pseudo R-Square for the Mugroso watershed for the coarsest domain and lead times up to 12 hours.







**Figure 4.** Accuracy Measures for deterministic precipitation and ensemble mean obtained from the WRF model . Domain 1 , domain 2 and domain 3 correspond respectively to the domains with resolutions 15 km, 5 km and 1.67 km.




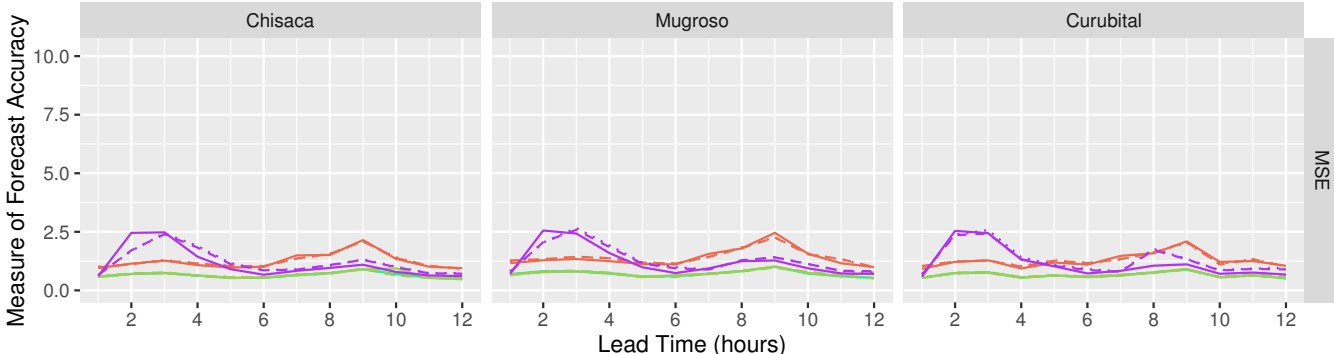

**Figure 5.** AAccuracy Measures for deterministic precipitation and ensemble mean obtained from the WRF model without the highest precipitation missed by the WRF model. Domain 1 , domain 2 and domain 3 correspond respectively to the domains with resolutions 15 km, 5 km and 1.67 km. The legend is the same as in Figure 4.

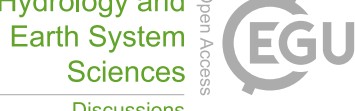



**Figure 6.** Performance Measures for deterministic discharge and ensemble mean forecasts. Domain 1 , domain 2 and domain 3 correspond respectively to the domains with resolutions 15 km, 5 km and 1.67 km.





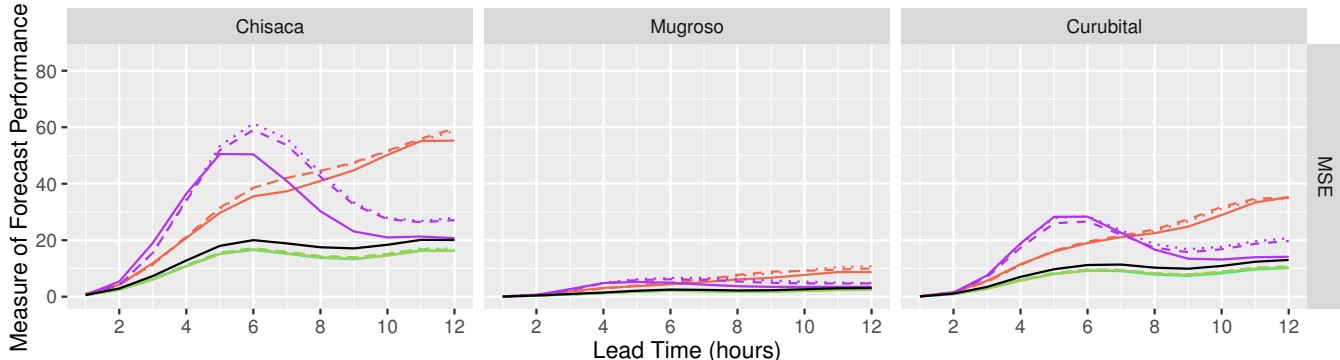

**Figure 7.** Performance Measures for deterministic discharge and ensemble mean forecasts obtained excluding the storm with the highest precipitation missed by the WRF model. Domain 1 , domain 2 and domain 3 correspond respectively to the domains with resolutions 15 km, 5 km and 1.67 km. The legend is the same as in Figure 6.

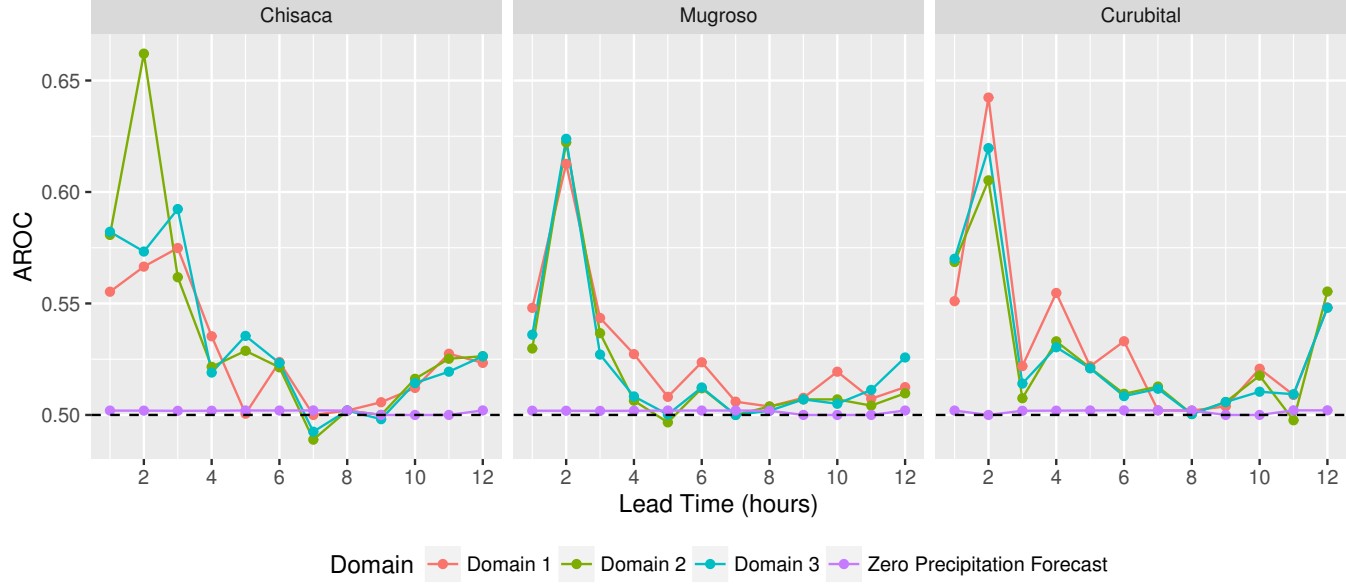

**Figure 8.** Area under the ROC curve (AROC) for the forecast of occurrence of precipitation with the probit model. Domain 1 , domain 2 and domain 3 correspond respectively to the domains with resolutions 15 km, 5 km and 1.67 km.





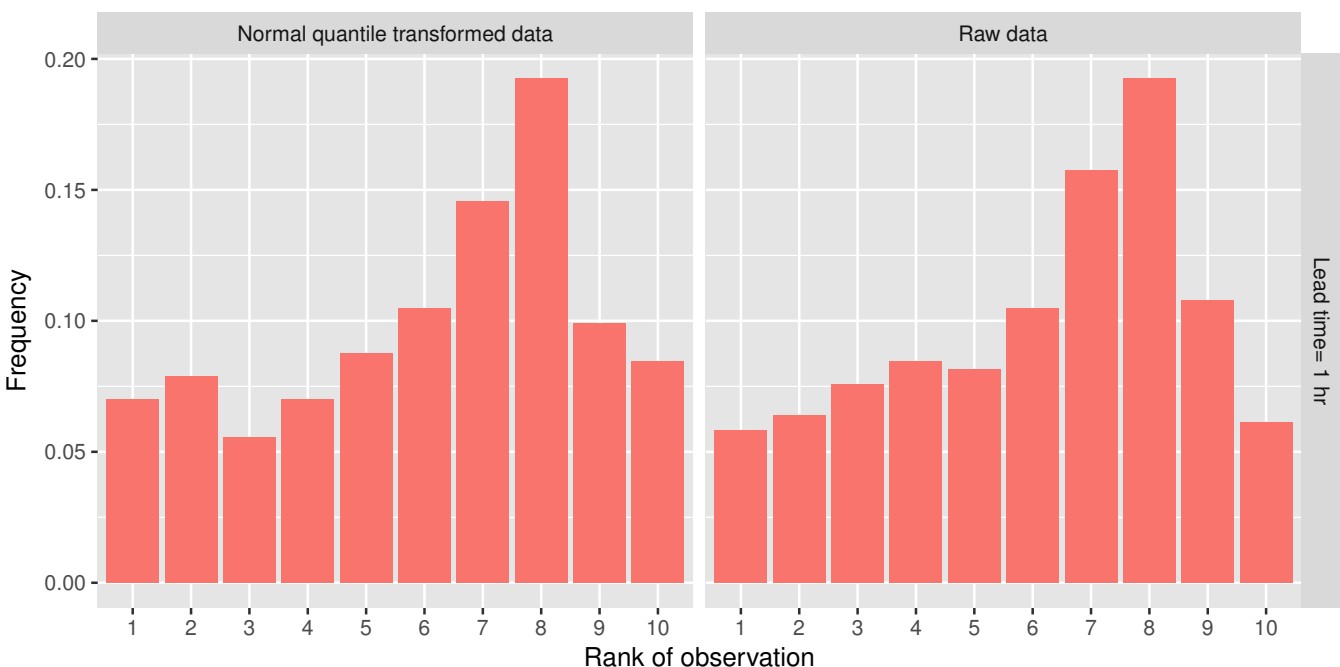

**Figure 9.** Rank histograms for the WRF ensemble for the Mugroso watershed for the finest domain (resolution of 1.67 km) and lead times up to 6 hours.

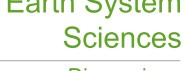
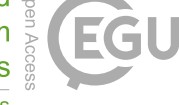


**Figure 10.** Rank histograms for the Q ensemble using as reference time series the discharge simulated with the TOPMODEL.





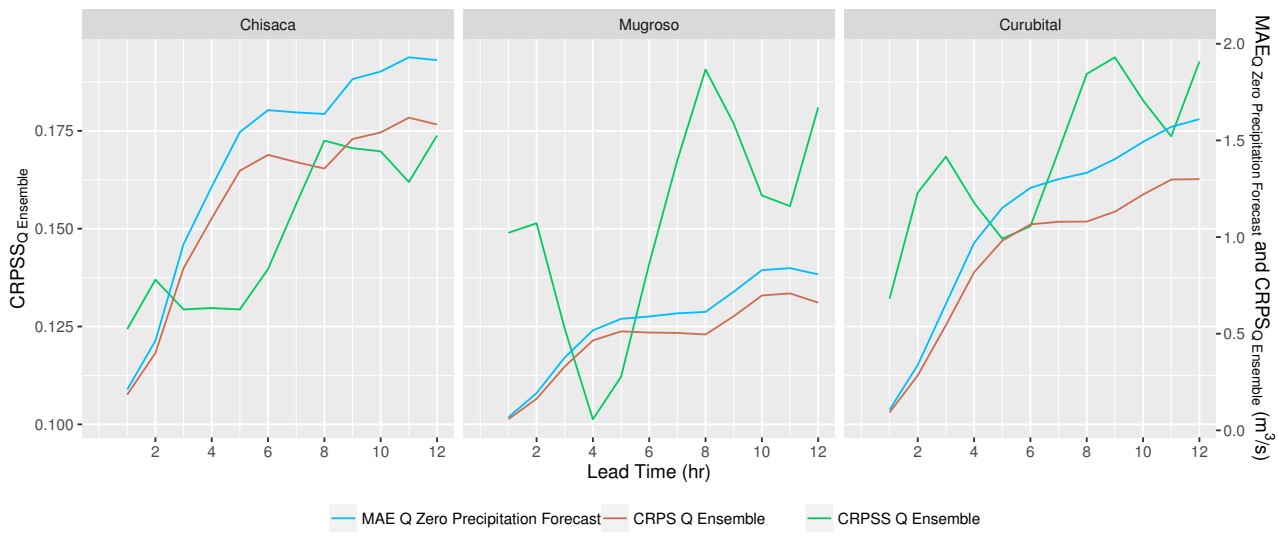

**Figure 11.** CRPSS for the discharge ensembles.





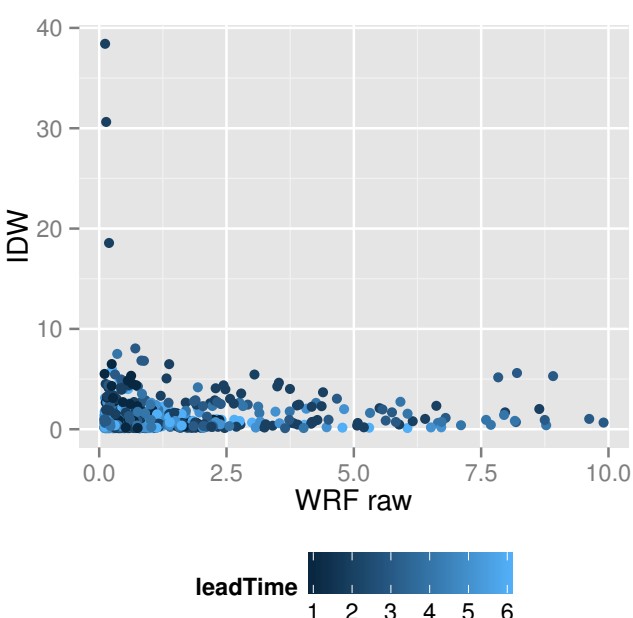

**Figure 12.** IDW precipitation for the Chisaca, Mugroso and Curubital watersheds vs WRF raw precipitation for the highest resolution domain (1.67 km).