# Peer review of "Streamflow forecasts from WRF precipitation for flood early warning in mountain tropical areas"

_Hydrology and Earth System Sciences, 2017_

## Referee Comment (RC1) · Anonymous Referee #1 · 8 Sep 2017

Overall, convincing case that finer spatial resolution and bias correction does not improve forecast skill of heavy precipitation in regions of tropical convection. However, statistical post processing to create a probabilistic forecast does enhance the ability to forecast a probability of precipitation at short lead times.

Given this, I recommend more discussion from a physical perspective of how the probit model method derives it's skill. It seems that this is the first stage (probit model) creates the biggest value in post-processing, rather than the second stage (quantile regression); it may be worth evaluating that.

p4 l20: I infer that you do not use a NWP ensemble (either based on GPS or WRF),
but a single "deterministic" GPS->WRF forecast. That should be emphasized here for clarity.

p4 l35: For clarity, reflect the language on p2 l32 here to keep clear the distinction between DBS and probit post processing.

p5 l10: The word "pre-processing" is unclear here, as it suggests *before* running WRF; do you mean post-processing?

p5 l13: As reference values the *observed* (?) time series of mean areal precip

p5 l20: Need more detail here: what periods were used in WRF and OBS to generate the gamma functions? Just the 107 storm days? This is relevant later to the extent the same periods are used for training and evaluation.

p5 l25: A more qualitative and physically based description needed here that does not depend on the reader fully understanding the statistical method. Is information from the neighborhood used for probability to form an ensemble? Below, what are the x covariates physically?

p8 l10: Why include SSclim and SS0 if not discussed?

p9 l21: Computed from the 107 selected storm days? Are the CDFs for the same days as used to train the bias correction?

p10: delete empty section 3.3

p10 l5 *shown* in Eqn 3 to Eqn 14

p12 l18: I recommend deleting fig 10 since it is barely discussed and expanding discussion of Fig 11.

p13 l22: Fig 12 doesn't show this... Is there a missing figure?

---

## Referee Comment (RC2) · Anonymous Referee #2 · 25 Sep 2017

Overall, the manuscript titled 'Streamflow forecasts from WRF precipitation for flood early warning in mountain tropical areas' demonstrates clear knowledge regarding numerical weather prediction modelling, particularly precipitation, and validation against observation values. Their analysis using raw model output, post-processed output, and ensemble data, shows that WRF forecasts can add additional skill when predicting precipitation in mountainous areas which could result in flooding. While the gain in skill may be relatively minor, as the authors discuss, the additional skill is forwarded onto any hydrological models, and would result in higher accuracy regarding flood timing and intensity.

A few further additions to the manuscript would be beneficial:

1) The simulation uses 3 domains- please make it clear, perhaps on Figure 1, the boundaries of the three domains.

2) Also indicate the relationship between the domains, i.e. is there two-way nesting to allow communication between domains or does the communication only go one way.

3) If communication is in fact two directions, please add discussion why results are different between the three domains

4) Model settings: The authors refer to another manuscript in regards to their model setup. It would be helpful if they listed a few of the major settings of the model in this manuscript, primarily choice of cumulus parameterization setting as well as micro-physics setting.

5) The finest domain is 1.67 km, which is well within the grid-spacing necessary to adequately resolve convective precipitation. Please add discussion in this regard, as well as whenever it is mentioned that the model poorly resolved convective precipitation in the discussion.

6) References: there are numerous little errors within the reference list that could use correcting.

Minor comments:

P1,L6: WRF acronym used in abstract without its definition

P3, L9: Paramo, is this supposed to be capitalized or not. Not consistent throughout introduction

P4, L11: Please indicate the date range where the 107 selected days come from. MSE/MAE/ME equations come well after the first acronym is used, P5 L9

P11 L33: this one sentence paragraph can be added into the previous paragraph

P12 L27: a comma ( , ) is surrounded by white space

P13 L1: 'Lead limes' should become 'lead times'

Figures/Tables:

Figure 1: include domain structure into this figure somehow.

Figure 2: "Precipitawon" shows up in the left side, strange formatting with this word too

Figure 3: Please describe what 'Q-Q' means within the manuscript and this figure caption

Figures 6-7: While all use the same legend style, the readers should not have to flip from one figure to the next to determine what each line means. Please include the legend on each figure.

---

## Author Comment (AC1) · 23 Nov 2017

**Response to Referee No 1**

We would like to thank the anonymous referee for an extensive review of our

manuscript and for providing us with helpful and constructive comments. For ease of reading we have copied the reviewer comments, as well as our response. We have also attached a track changes version of the manuscript, using as original version the one including the modifications suggested by the Editor.

**GENERAL COMMENTS**

**Overall, convincing case that finer spatial resolution and bias correction does not improve forecast skill of heavy precipitation in regions of tropical convection. However, statistical post processing to create a probabilistic forecast does enhance the ability to forecast a probability of precipitation at short lead times. Given this, I recommend more discussion from a physical perspective of how the probit model method derives it's skill. It seems that this is the first stage (probit model) creates the biggest value in post-processing, rather than the second stage (quantile regression); it may be worth evaluating that.**
RESPONSE:
We thank the reviewer for this suggestion. We extended the discussion in section 4.1 (Evaluating precipitation forecasts from the WRF model) as follows:
*The AROC of the occurrence of precipitation shows that values higher than 0.5 are obtained in all watersheds for lead times 10 up to 5 hours, indicating that the probit model has the ability to discriminate between the occurrence or non-occurrence of precipitation up to these lead times. In all domains and watersheds, the highest values of AROC are obtained for lead-times between 2 and 3 hours, in the range of 0.63-0.67. This means that there is some skill in the probit model to forecast occurrence of precipitation for lead times up to 5 hours, even if the skill of the amount of precipitation is low. For longer times the probit model does not have any skill, consistent with other studies where the performance significantly reduces after a few hours (Liu et al., 2015). This behaviour of the probit model implies that for lead times up to 5 hours the*

*probability of occurrence of precipitation can be modelled from the WRF precipitation with some skill, meaning that the higher the WRF forecast precipitation value the higher the probability of occurrence of precipitation. However, even though the lead times at which the model still has skill are limited, this does imply that the current ability of NWP in tropical mountainous areas can provide a limited extension of lead time beyond the concentration time of small, flashy, watersheds.*

**SPECIFIC COMMENTS**

1. **p4 l20: I infer that you do not use a NWP ensemble (either based on GPS or WRF), but a single "deterministic" GPS-WRF forecast. That should be emphasized here for clarity.**
   RESPONSE: We thank the reviewer for this contribution. The clarification was added in section 2.2, as follows:

   *The dataset of WRF forecast used in this paper corresponds to 107 selected days when significant storms were recorded, during the period July 2009-December 2002. For each of these days, deterministic forecasts with the WRF model were generated at 00:00 GTM, 06:00 GTM, 12:00 GTM and 18:00 GTM.*

2. **p4 l35: For clarity, reflect the language on p2 l32 here to keep clear the distinction between DBS and probit post processing.**
   RESPONSE: The paragraph was modified as follows:

   *From T0 the model was forced using four different rainfall forecasts; a) zero rainfall forecasts; b) raw forecasts from the WRF model; c) bias corrected WRF forecasts; d) and precipitation forecast ensembles obtained from the post processing of the WRF model, based on probit regression and quantile regression.*
3. **p5 l10: The word "pre-processing" is unclear here, as it suggests \*before\* running WRF; do you mean post-processing?**
RESPONSE: We thank the reviewer for pointing out this mistake. The paragraph was modified as follows:

*(b) Raw forecasts from the WRF: the only post-processing of the WRF forecasts is the sampling of the grids to obtain the hourly mean areal precipitation for each watershed.*

4. **p5 l13: As reference values the \*observed\* (?) time series of mean areal precip**
RESPONSE: The word \*observed\* was included, as follows:

*As reference values the observed time series of mean areal precipitation obtained from IDW are used, and the correction is carried out for each lead-time of the WRF model.*

5. **p5 l20: Need more detail here: what periods were used in WRF and OBS to generate the gamma functions? Just the 107 storm days? This is relevant later to the extent the same periods are used for training and evaluation.**
RESPONSE: A clarification was added as follows:

*(c) Bias correction of WRF: The time series of mean areal precipitation obtained from the WRF model are bias corrected through Distribution-Based Scaling - DBS (Yang et al., 2010). Mean areal precipitation was established using forecasts made for the 107 selected storm days as the biases during more extreme events is the prime interest.*

6. **p5 l25: A more qualitative and physically based description needed here that does not depend on the reader fully understanding the statistical method. Is information from the neighborhood used for probability to form an ensemble? Below, what are the x covariates physically?**

RESPONSE:

The explanation on the bias correction and generation of ensembles through post processing of the rainfall forecast from the WRF model, was improved as follows:

*(d). Bias correction and generation of ensembles through post processing of the rainfall forecast from the WRF model: The reference and forecast time series were organised according to lead time and a two stage post processing model was applied to reflect the intermittent nature of rainfall (Rene et al., 2012; Clark et al., 2004). The first stage corresponds to the probit model (Scardovi, 2015; Kleiber et al., 2012; Bremnes, 2004) to simulate the probability of occurrence of precipitation. The second stage considers the probability distribution of the precipitation amount, given its occurrence, for which quantile regression was used. The mean precipitation time series (observed hourly mean areal precipitation for each watershed) is first disaggregated into a time series of occurrence (1 = wet time step and 0= dry time step) and precipitation amounts, which is established for the wet time steps only. The time series of occurrence is used as the response variable for the probit regression model. The probit model allows the probability of occurrence of precipitation to be estimated using the WRF hourly mean areal precipitation over each watershed as a predictor or secondary variable. The time series of precipitation amounts is used as the response variable for the quantile regression model. This allows different quantiles of the response variable to be estimated using the WRF hourly mean precipitation for each watershed as predictor or secondary variable.*

7. **p8 l10: Why include SSclim and SS0 if not discussed?**
   RESPONSE:
   We believe that SSclim and SS0 are very important in the results and discussion sections. The results of these two parameters are presented and explained in section 3.4, -in the case of precipitation-, and in section 3.5 in the case of discharge. The behaviour of SSclim and SS0 supports part of the discussion

in section 4.1. In section 4.2 SSclim supports the discussion on lead times, at which the added value is totally lost in comparison with the climatology. In order to clarify the importance of these parameters, section 4.1 was complemented as follows:

*The WRF data used in this study shows limitations in fulfilling both these conditions, which leads to the post-processed precipitation being no more skilful than the sample climatology (as shown by negative values of SSclim) and providing only a modest improvement in comparison to the zero precipitation forecast (as shown by most values of SS0 being positive, albeit low).*

8. **p9 l21: Computed from the 107 selected storm days? Are the CDFs for the same days as used to train the bias correction?**
RESPONSE:
The CDFs were changed to Q-Q plots for clarity, attending a comment from the Editor suggesting a clearer figure. A clarification was added that the figures correspond to the 107 selected storms. Additionally, in the methodology, a clarification was included in section 3.1 and 2.3.1. The modifications are as follows:

*Section 3.1:*
*Figure 3 shows the quantile-quantile plots (Q-Q plots), comparing the WRF precipitation and the bias corrected WRF precipitation with the observed precipitation obtained from IDW, for the 107 selected days of significant precipitation.*

In addition, a clarification was made in section 2.3.1 in response also to comment 5.

9. **p10: delete empty section 3.3.**
RESPONSE:

We thank the reviewer for pointing out this mistake. Empty section 3.3 was deleted.

10. **p10 l5 *shown* in Eqn 3 to Eqn 14**
RESPONSE:
We thank the reviewer for pointing out this mistake. The word was corrected.

11. **11. p12 l18: I recommend deleting fig 10 since it is barely discussed and expanding discussion of Fig 11.**
RESPONSE:
We believe that Figure 10 (Figure 11 in the attached version of the manuscript) is important for our discussion. The figure is presented in section 3.5. However, in section 4.2 (Evaluating discharge forecast the rank histograms are discussed), the figure is not mentioned explicitly, we believe that this may be the reason why the importance of the figure is not clear. We included a reference to the figure in section 4.2, as follows:

*The hydrological model produces ensemble results with rank histograms that do not reflect the overprediction of the precipitation ensemble, with approximately uniform rank histograms mainly for the first lead times (see Figure 11). Approximately the same shape of the rank histograms is observed for ensembles obtained from quantile regression when no transformation is used in the precipitation data and when raw data is used. The minor influence of the transformation is consistent in all the performance assessments. This may be due to the relatively low improvement of the fit of the quantile regressions when the normal quantile transformation is used. According to Figure 4, the most significant improvement is found in the low percentiles (up to the 25th percentile). For higher percentiles, the improvement is not significant and for some percentiles degradation of goodness of fit occurs. As floods occur for higher rainfall percentiles, the similarity in behaviour is logical.*

Regarding Figure 11 (Figure 12 in the attached version of the manuscript) the discussion is included in section 4.1. However, the figure is also not mentioned explicitly in the text of the discussion. The text in the discussion was complemented as follows:

*The results of the CRPSS (see Figure 12) show that forcing the hydrological model with WRF ensembles improves the forecasts in a range of 8.5-22% in comparison with a forecast produced with the hydrological model forced with zero precipitation for lead times between 1 and 12 hours. The positive values of the CRPSS imply added value to the forecasts, albeit modest. The CRPSS obtained from the ensembles is comparable to the CRPSS values found in other areas with other NWP models, albeit in the low range. E.g. Robertson et al. (2013) found a CRPSS of 37% on average for post processed ensembles in Australia, where rainfall is predominantly produced by large-scale synoptic systems that are better predicted by NWP models. Therefore, given the high complexity of the meteorological conditions of the study area, and despite the relatively poor skill of the WRF model in predicting precipitation amounts, the WRF model does show promise at producing a benefit in its use for flood forecasting compared to not using precipitation forecasts. This is likely due to the skill found in the probit model in predicting the occurrence of rainfall.*

12. **12. p13 l22: Fig 12 doesn't show this. . . Is there a missing figure?**
RESPONSE:
We agree with the reviewer that the figure showing IDW precipitation vs WRF raw precipitation does not show the reduction of differences between observations and bias corrected WRF precipitation. Figure 12 was eliminated (this change was made attending a suggestion of the Editor) and was replaced in this paragraph by Figure 3 (please see attached version of the manuscript) that shows a Q-Q plot of the observed IDW precipitation vs WRF and WRF bias corrected precipitation, showing clearly the effect of the bias correction.

**Supplement:**

[revised manuscript text omitted]

---

## Author Comment (AC2) · 23 Nov 2017

**Response to Referee No 2**

We would like to thank the anonymous referee for the thorough and constructive

review of the manuscript. We have carefully considered the comments, and through this document we would like to provide a detailed response to each, as well as how we have adapted the manuscript where applicable. We have also attached a track changes version of the manuscript.

**SPECIFIC COMMENTS**

1. **The simulation uses 3 domains- please make it clear, perhaps on Figure 1, the boundaries of the three domains.**
   RESPONSE:
   We thank the reviewer for this suggestion. The domains were included in Figure 1.

2. **Also indicate the relationship between the domains, i.e. is there two-way nesting to allow communication between domains or does the communication only go one way.**
   RESPONSE:
   We thank the reviewer for pointing out the need for clarification on the relationship between domains. In the description of the WRF model (section 2.2 WRF model data and observed rainfall fields), the information was added as follows:

   *The WRF model comprises three nested domains, centred in BogotaÌĄ. The coarsest domain covers most of the Colombian National territory with a spatial resolution of 15 km; the intermediate domain covers mainly the central and eastern Andean cordilleras with a spatial resolution of 5 km; and the finest domain covering BogotaÌĄ and the study area only has a spatial resolution of 1.67 km (Arango and Ruiz, 2011). The nested model domains have been set up using two-way communication (without smoothing).*

3. **If communication is in fact two directions, please add discussion why re-
sults are different between the three domains**
RESPONSE:
In subsection 4.1 (Evaluating precipitation forecasts from the WRF model), the
following paragraph was extended:

*No significant differences among three domains of the WRF model were found
in the behaviour of the scores for the tests carried out in this study. The two-way
communication between the nested domains implies that precipitation volumes
are aligned between the different resolutions, with the small differences found
likely being due to differing ratios between the WRF model grid cell sizes and the
sizes of the watersheds*

4. **Model settings: The authors refer to another manuscript in regards to their
model setup. It would be helpful if they listed a few of the major settings of
the model in this manuscript, primarily choice of cumulus parameterization
setting as well as microphysics setting.**
RESPONSE:
We agree that adding the major settings improves clarity. The major settings of
the model were included in section 2.2 (WRF model data and observed rainfall
fields), as follows:

*The parameterisation of the model corresponds to that used by IDEAM for its
routine forecasts (Arango and Ruiz, 2011), using the Kain-Fritsh cumulus param-
eterisation scheme, except for the finest domain where convection is not parame-
terised but resolved. Microphysics is parameterised as the WRF Single-Moment
3-class scheme (WSM3).*

5. **The finest domain is 1.67 km, which is well within the grid-spacing neces-
sary to adequately resolve convective precipitation. Please add discussion
in this regard, as well as whenever it is mentioned that the model poorly**

**resolved convective precipitation in the discussion**
RESPONSE:
In section 4.1 we discussed the issue of resolution and we added an explanation of the scope of the paper:

*...Other applications of NWP (e.g. Roberts et al. (2009)) show that finer resolutions are capable of producing more accurate predictions, and that physics configuration, resolution and initial conditions highly influence the WRF model performance (Kryza et al., 2013). The similarity of results regardless of resolution found in this study may be also related to deficiencies in the parameterisation of the model, or to the inability to sufficiently resolve the topography. A more detailed review of the WRF model would be required to reveal possible deficiencies, and to suggest improvements....*
*The WRF model has been shown to be highly sensitive to the parameterisation of cumulus and microphysical processes (Remesan et al., 2014; Rama Rao et al., 2012). Parameter sensitivity is generally dependent on local conditions (Di et al., 2014), and the best configuration varies with time and rainfall threshold (Jankov et al., 2005). While it was our objective to test the configuration of the WRF model as it is used operationally by IDEAM and optimitising the parameteriszation of the WRF model is outside the scope of the present paper, further work could test various model parameterisations, assessing their sensitivity and searching for an optimal or a set of optimal configurations that help to improve parameterisation (in the case of the coarser model domains) and in particular resolving (in the case of the finest model domain) of convective precipitation as frequently occurs in the case study area.*

6. **References: there are numerous little errors within the reference list that could use correcting.**
RESPONSE:
The references were reviewed and corrected.

[Figure]

7. **P1,L6: WRF acronym used in abstract without its definition**
   RESPONSE:
   We added the meaning of the acronym to the abstract.

8. **P3, L9: Paramo, is this supposed to be capitalized or not. Not consistent throughout introduction**
   RESPONSE:
   The capital P was replaced by p.

9. **P4, L11: Please indicate the date range where the 107 selected days come from.**
   RESPONSE:
   The period was included as follows:
   *The dataset of WRF forecast used in this paper corresponds to 107 selected days when significant storms were recorded, during the period July 2009-December 2002*

10. **MSE/MAE/ME equations come well after the first acronym is used, P5 L9**
    RESPONSE:
    We thank the reviewer for this comment; indeed the acronyms are used in a section before the equations are presented. However, when the acronyms are first used, it is made in the context of the introduction of the methodology (summary of the methodological approach) and the corresponding meaning of the acronym is used. After that, the acronyms are used to introduce the equations. We believe that this use of the acronyms is clear and there is no need to modify the text.

11. **P11 L33: this one sentence paragraph can be added into the previous paragraph**
    RESPONSE:
    The sentence was moved to the previous paragraph

12. **P12 L27: a comma ( , ) is surrounded by white space**
RESPONSE:
The white space was deleted

13. **P13 L1: 'Lead limes' should become 'lead times'**
RESPONSE:
The word was corrected

14. **Figure 1: include domain structure into this figure somehow.**
RESPONSE:
The domains were included in Figure 1

15. **Figure 2: "Precipitawon" shows up in the left side, strange formatting with this word too**
RESPONSE:
The error and formatting was corrected.

16. **Figure 3: Please describe what 'Q-Q' means within the manuscript and this figure caption**
RESPONSE:
In the manuscript the following sentence was added:
*Figure 3 shows the quantile-quantile plots (Q-Q plots) comparing the WRF precipitation and the bias corrected WRF precipitation with the observed precipitation obtained from IDW.*
The caption of the figure was changed to:
*Q-Q plots (quantile-quantile plot, plot of the quantiles of the first data set against the quantiles of the second data set) for the Mugroso watershed comparing observed precipitation with WRF and WRF bias corrected precipitation*

17. **Figures 6-7: While all use the same legend style, the readers should not have to flip from one figure to the next to determine what each line means.**
**Please include the legend on each figure.**
RESPONSE:
The legend was added to figures 6-7.

**Supplement:**

[revised manuscript text omitted]